# Self-Reported Household Waste Recycling and Segregation Practices among Families in Eastern Region of Saudi Arabia: A Cross-Sectional Study

**DOI:** 10.3390/ijerph20031790

**Published:** 2023-01-18

**Authors:** Yousif Mohammed Elmosaad, Ahmed M. Al Rajeh, Maria Blesilda B. Llaguno, Sami Saad Alqaimi, Ali Mohammed Alsalman, Ali Yousif Alkishi, Hassan Hussain, Mohammed Ahmed Alhoudaib, Othman Saad Alnajim, Safia Belal

**Affiliations:** 1Department of Public Health, College of Applied Medical Sciences, King Faisal University, Al Ahsa 37912, Saudi Arabia; 2Department of Respiratory Care, College of Applied Medical Sciences, King Faisal University, Al Ahsa 37912, Saudi Arabia; 3Department of Nursing, College of Applied Medical Sciences, King Faisal University, Al Ahsa 37912, Saudi Arabia; 4Department of Health Informatics, College of Applied Medical Sciences, King Faisal University, Al Ahsa 37912, Saudi Arabia

**Keywords:** awareness, practices, waste management, waste recycling and waste segregation

## Abstract

Background: The reuse and recycling of household waste are correlated with a household’s daily activities and commonly depend on sociodemographic factors. In this study, we aimed to assess and probe the level of awareness of waste reuse and recycling, self-reported household waste recycling and segregation practices, and the variables affecting the practices of households in Al-Ahsa, Saudi Arabia. Methods: We employed a cross-sectional study utilizing the multi-stage random sampling of 279 households and a researcher-structured, online questionnaire in English and Arabic. Data were analyzed using SPSS Version 20. Descriptive statistics was used to describe the level of awareness/practices, and inferential statistics was used to describe the correlational aspects. Results: It was determined that female participants, younger participants, participants of both genders with university and postgraduate education levels, and urban residents were significantly associated with self-reported household waste segregation and recycling practices at the source. Plastics, paper, glass, food waste, textiles, and electronic waste were determined to be the most common types of household waste. A lack of expertise, awareness, demand for recycled products, and laws that support recycling was reported to affect female participants’ failure to recycle. Social media, television, and educational institutions were shown to be sources of information regarding waste segregation and recycling. Therefore, awareness-raising polices must be developed to improve the prevalence, expertise, and efficiency regarding recycling and segregation. In addition, inventive methods, such as a card-based reward system, should be used to increase the demand level for recycled products.

## 1. Introduction

Worldwide, municipal solid waste (MSW) production was estimated to be 2.01 billion metric tons (MT) in 2016 and is expected to increase by 70% in 2050 due to rapid population growth, economic development, and urbanization [1,2]. The composition of this waste significantly varies from country to country and is mainly governed by the lifestyle, economic situation [3], population growth, and consumer shopping habits within each country [4]. This waste mainly consists of paper, plastic, and organic waste [1]. In developing countries, over 90% of waste is often disposed in unregulated dumps or is openly burned, thereby imposing significant health risks to humans and the environment [5,6].

In the case of countries in the Gulf Cooperation Council (GCC), the total amount of annually generated solid waste is estimated to be between 95 and 100 million MT [7]. This high figure is due to the fact that in Saudi Arabia, Bahrain, the UAE, Qatar, and Kuwait, the global waste generation average of 1.2 kg/person/day is exceeded. The highest amount of waste is generated per person per day in the UAE, at 2.1 kg, followed by Saudi Arabia, Qatar, and Bahrain, all at 1.7 kg/person/day [8].

Saudi Arabia is the largest country in terms of population in the Persian Gulf region, with a population size of 33.7 million [9]. In addition to rapid economic development and industrialization, fast urbanization also occurred in Saudi Arabia, which resulted in the generation of large amounts of waste in rural and urban areas, which in turn contributes to increases in waste and pollution levels. The Kingdom produces 15 million tons of MSW each year, with an average daily rate of 1.4 kg per person, and this figure is expected to increase to 30 MT each year by 2033 with an estimated per capita waste generation of 2.9 kg/person/day [10]. The current MSW management system in the KSA is simple: collection, dumping in landfill sites, and combustion. This practice means that the greenhouse gas (GHG) emissions (CO_2_, CH_4_, and N^2^O) are the fourth highest polluters after the levels of fossil fuels [11,12], leachate production, and soil contamination [13].

The location of this study, Al-Ahsa, Saudi Arabia, has a population of 1,200,000 residents, who generate approximately 3800 m^3^ of waste per day [14]. The average daily rate is 0.95 kg/capita/day, and the waste mainly comprises paper and cardboard, food, plastics, wood, metals, and glass (17.09%, 14.73%, 13.81%, 13.51%, 11.41%, and 10.82%, respectively) [15]. Currently, most MSW is disposed of in landfill sites without any treatment, with only a small amount being recycled [14].

Recognizing the hazards related to uncontrolled dumping, the Saudi government approved new regulations to ensure an integrated framework for the management of municipal waste through the introduction of modern waste management techniques. These measures were intended to significantly increase waste recycling and reuse from 15% to 40%. These measures were also intended to generate business opportunities [10] and increase environmental sustainability [13].

A review of the current practices and future opportunities that have been adopted for solid waste collection, handling, and disposal in Saudi Arabia was conducted in a similar study which reported that the MSW generated in the Kingdom can lead to contrasting consequences: either environmental/health hazards or wealth generation. Consequently, the authors of that study proposed that a reversal approach for MSW management be adopted to produce less MSW. This strategy advocates the prioritization of source reduction, a continuous focus on waste minimization practices through education and electronic media, and the promotion of the 3Rs (Reduce, Recycle, and Reuse) approach [15]. It is noteworthy that the 3Rs approach can also be applied at the source of waste generation. Furthermore, it was suggested that strong rules and heightened information dissemination initiatives be implemented. A significant increase in the private sector’s involvement in the process of organic, plastic, and metal waste collection for recycling and reuse was likewise noted in another study [16].

Another study sought to probe the ongoing trends of waste generation, determine the best waste management techniques, and provide methods to improve the foundation of SWM within the Kingdom. After analyzing the MSW generated in the region and examining the current and potential waste handling schemes of the Kingdom, an institutional framework for Integrated Solid Waste Management (ISWM) was designed by the authors. ISWM is the strategic approach for the sustainable management of solid waste covering all sources and aspects such as generation, segregation, sorting, transfer, treatment, recovery, and disposal in an integrated manner, emphasizing the maximization of efficiency [17]. Solid Waste Management (SWM), on the other hand, works on the principle of the 4Rs—Reduce, Reuse, Recycle, and Recover. Another waste management technique is Waste-to-Energy (WTE), in which energy is recovered from waste. Recovery is achieved by treating non-recyclable waste to generate energy that can be included in the energy production of a country [18]. The future implementation of WTE is promising in the Kingdom of Saudi Arabia due to Vision 2030’s ambitious measures to reduce the Kingdom’s dependency on oil and diversify its economy.

Recently, some efforts were likewise implemented in other countries to improve the management of MSW through environmental policies to drive the waste management strategy in a direction of waste recovery, waste reuse, and healthy disposal practices [19]. A significant increase in private sectors’ involvement in the process of organic, plastic, and metal waste collection for recycling and reuse [16] was also noted. In fact, recycling and reuse are noteworthy components of sustainable waste management, as these processes recover useful raw materials and energy and reduce waste pollution [20,21].

In essence, the efficiency of MSW recycling and reuse depends upon sufficient public awareness [22]. This is because a household’s awareness of waste reuse and recycling is closely related to the daily activities of its inhabitants, which commonly depend on sociodemographic factors [23]. Previous studies conducted worldwide have shown that waste generation is correlated with an individual’s sociodemographic variables [21,23,24]. Likewise, social influence significantly predicts households’ willingness to sort and recycle, that is, to promote recycling. It is noteworthy that one’s attitude, social influence, perceived behavioral control, market incentives, government facilitators, and awareness positively and significantly affect residents’ waste-sorting intentions [25]. In fact, these findings are consistent with the structural equation model, which shows that perceived behavioral control significantly predicts households’ willingness to sort and recycle waste.

The moderating effect of income and market incentives was likewise observed between attitude and willingness to sort and recycle waste in the low-income-level and high-income-level groups. In addition, the gender status of the participants had a moderating effect on the relationship between market incentives and willingness to sort and recycle waste in both males and females [26].

Therefore, most of the previous studies conducted in the KSA at the national and local levels only concentrated on exploring the correlation between socio-economic factors and MSW generation but ignored the direct and indirect effects of a household’s awareness of practicing waste recycling and segregation. In this study, the researchers aimed to assess households’ awareness of the recycling and segregation of waste and probe deeper into the variables affecting their practices. The extent of the association of a household’s demographic characteristics with their awareness level and practices of household waste recycling and segregation was likewise identified.

## 2. Materials and Methods

### 2.1. Study Design and Setting

A cross-sectional study was conducted among all the households in Al-ahsa province located in the eastern part of Saudi Arabia. This province covers 410,713 km^2^, which is divided into three administrative areas: Al-Hofuf, Al-Mubaraz, and Al-Gourah.

### 2.2. Study Population and Sampling

The estimated population of the province is 1,041,863 according to the Central Authority for Statistics, of which 877,845 are aged 18 years or over. The required study sample size was 381. This was determined through the stratified proportional allocation random sampling method using Epi Info software (with an absolute precision of 5% and at a 95% confidence interval). A total of 279 households responded to the survey questionnaire with a 73.2% response rate. A multistage random sampling technique was used to select a representative sample size from the study areas of Al-Hofuf, Al-Mubaraz, and Al-Gourah.

### 2.3. Instrument

The questionnaire was composed of 4 parts. Part one included variables regarding the household’s socio-demographic characteristics, such as education level, occupation, monthly family income, age, gender, and family size. Part two of the questionnaire comprised six questions, which were used to assess the household’s general knowledge about waste management.

The items in part three, collated from different sources, focused on the household’s awareness and practices regarding the segregation and recycling of household waste. The household awareness was assessed using a tool with 7-item questions rated using a 4-point Likert-type scale. In this scale, zero points were assigned to each item(s) answerable by either disagree or strongly disagree, and one point was assigned to each agree or strongly agree answer. The maximum possible score was seven points, with a range from zero to seven points. We assessed the participants’ waste segregation and recycling practices using Yes/No questions. The participants were granted one point for each activity they always practiced and zero points if they did not practice it. The maximum possible score was seven points, with a range from zero to seven points.

In terms of both awareness and practice, the mean score was used to classify the level of awareness and practice. When the level of each was equal to or higher than the mean cutoff point, it was considered that the households had good awareness and practices and vice versa.

Part 4 of the questionnaire requested information regarding types of waste, reasons for not recycling waste materials, and sources of information regarding waste recycling and segregation.

To ensure the clarity and relevance of the questions and to determine the time required to answer all of the items, the questionnaire was pilot tested among a simple random sample of households to check its format, language, sequence, comprehension of the questions, and duration were suitable. The reliability and validity of the tool was checked using Cronbach’s alpha test, the result of which was 0.81. Informed consent was obtained from each respondent, and the participants were informed about the purpose of the study, the confidentiality and privacy of the data gathered, and that the data were only used for the intended purposes of this study.

### 2.4. Data Collection

Data were collected from December 2021 to February 2022 utilizing an online, self-administered, anonymous, researcher-structured questionnaire (Google Forms) with both English and Arabic translations.

### 2.5. Statistical Analysis

The data were analyzed using the Statistical Package for Social Sciences (SPSS) version 20. Depending on the nature of the variables, descriptive statistics was used to tabulate and describe the data, e.g., the means were calculated for continuous variables and frequency, and percentages were calculated for categorical variables. Inferential statistics such as the chi-square test were used to determine the association between awareness and the sociodemographic background of the household and waste recycling and segregation. Binary logistic regression was used to identify the probable predictors of practicing waste segregation and recycling at the source, adjusted by the household’s level of awareness. All statistical analyses was set at a 95% confidence interval (CI) and a *p* < 0.05 level of significance. To determine the level of awareness and practice among households, a scoring system was applied: one point for each correct answer and zero points for an incorrect answer.

## 3. Results

### 3.1. Socio-Demographic Characteristics of the Study Participants

Table 1 shows the socio-demographic characteristics of the study participants. Out of the 279 participants, those working in the government (89 or 31.9%) outnumbered those who were unemployed (83 or 29.7%), while the number of self-employed participants was the lowest (30 or 10.8%). In terms of the household educational level, the majority (189 or 67.7%) were university-educated, followed by those who were high school students (46 or 16.5%), and the fewest were doctorate graduates (7 or 2.5%).

In terms of the household age group, the mean age of the respondents was 38.4 ± 13.1 years, with 80 (or 28.7%) falling in the 38–47 age range, 74 (or 26.5%) between 18 and 27 years old, and 21 (or 7.5%) above 58 years old. With regard to the homemaker education level, our results showed that there were 105 (or 37.6%) high school graduates, 97 participants (or 34.8%) were college graduates, and 6 (or 2.2%) had postgraduate degrees. In terms of the household monthly income, half of the participants (141 or 50.5%) earned more than SAR 11,001, followed by 64 participants (or 22.9%) who earned SAR 9001 to 11,000; only 21 participants (or 7.5%) earned SAR 4000 to 6000. In terms of gender, the majority were female (195 or 69.9%), while only 84 (or 30.1%) were male. The average family size was 6.5 ± 2.4, with 144 (or 51.6%) participants having families with six members or less and 135 (or 48.4%) having seven family members or more.

### 3.2. General Information about Waste Management: Households’ Variations Based on Their Gender on Waste Management

The chi-square test was used to determine the association between waste management and gender. As shown in Table 2, there was a significant association between gender and waste management (*p* < 0.5). The majority of female respondents recognized waste as a problem in the KSA (72.8%), that waste collection services are available in their area (74.7%), that waste is collected every day (73.8%), that an act on waste exists (78.3%), and that waste management needs to be paid more attention to (68.1%).

### 3.3. Awareness about Waste Segregation and Recycling

Table 3 presents the participants’ responses related to their awareness of waste recycling. The results indicate that 60.9% of the study’s participants had a poor overall level of awareness about waste recycling. However, it is noteworthy that 87.1% of them agreed that waste recycling helps conserve the natural resources, 86.7% knew that recycling is the process of converting waste into new products, and 83.9% knew waste recycling saves money and energy. Moreover, 83.5% and 81.7% knew that waste recycling is a shared responsibility among family members and that by reducing the waste, we decrease the need for more landfill.

Regarding the association between selected socio-demographic variables and the overall level of awareness about waste recycling, the chi-square test results indicated that awareness was associated with gender, age group, education, and family size (*p* < 0.05).

### 3.4. Awareness of the Households about Waste Segregation

A set of statements was developed to measure the participants’ awareness of waste segregation, as seen in Table 4. The results showed that the majority (70.3%) of the study participants had a poor awareness level about waste segregation. However, approximately 88.5% of the study participants agreed that individuals and communities have roles and responsibilities regarding waste segregation, 87.5% knew that waste segregation at the household level can facilitate its reuse and recycling, and 93.5% knew that the effective segregation of waste means that less waste is deposited in landfill. Moreover, 53.0% of the respondents claimed that the segregation of household waste is time-consuming. When participants were asked about the availability of waste segregation containers near their houses, 80.3% of the study participants knew that waste segregation containers were not available near their houses.

Regarding the association between selected sociodemographic variables and the overall level of awareness about waste segregation, the chi-square test results indicate that awareness was associated with gender, age group, and education (*p* < 0.05); no association was noted with family size (*p* > 0.05).

### 3.5. Logistic Regression for Socio-Demographic Characteristics of the Household as Predictor of Practicing Waste Segregation and Recycling at Source Adjusted by Level of Knowledge

In the multivariate analysis, it is worth noting that some socio-demographic characteristics were significantly associated with self-reported household waste segregation and recycling at the source. These were age, gender, and the level of education of the male members of the household. Specifically, the age groups of 38–47, 48–57, and >58 years were determined to be less likely to recycle waste at the source (B = −1.63, −3.04 and −2.91, respectively) than younger households (18–27 years old). Additionally, it was shown that male and female household members had a good level of awareness about waste recycling at the source and were 2.75 times more likely to have had good self-reported waste recycling practice at the source than households who had a poor level of awareness (OR = 2.75, 95% CI: 1.23–6.16). However, family size, the level of education of the female members of the household, and household income were not associated with the self-reported household recycling of waste at the source (*p* > 0.05).

Both genders in this study showed significant associations with self-reported household waste segregation at the source. Specifically, female households were 2.35 times more likely to have good self-reported waste segregation practice at the source compared to their male counterparts (OR = 2.35, 95% CI: 1.65–2.79). Moreover, households of males who had university and postgraduate education were approximately 2.2 times more likely to have good self-reported segregation practices than households of males who had a high school degree or lower level of education (OR = 2.15, 95% CI: 1.35–3.81). Furthermore, households with good levels of awareness about waste segregation were 2.85 times more likely to have good self-reported waste segregation practice at the source compared with those who had poor levels of awareness (OR = 2.85, 95% CI: 1.45–5.62). The results further showed that households in a rural area (in Alqarah Saudi Arabia) were less likely to have had good self-reported waste segregation practices at the source than households in an urban area (Alhofuf) (B = −1.14, (OR = 0.32, 95% CI: 0.12–0.86)). However, household age, occupation, average monthly income, and family size were not associated with waste segregation practice, as seen in Table 5.

### 3.6. Type of Waste Generated by Household

In terms of the most common types of waste, it was reported that the waste generated by male and female households in the study was mostly plastic (45.5%). The remaining reported types of waste were paper (15.4%), glass (12.9%), food waste (10.1%), textiles (old clothes) (8.6%), and battery and electronic waste (3.9%). This is shown in Table 6.

### 3.7. Association between the Types of Waste and Gender

A chi-square test was used to identify the association between the types of waste and gender. The results showed a significant association between gender and waste types (*p* < 0.05). This indicated that female-headed households generated plastic, paper, glass, food, and textile waste more than male-headed households did. The most common type of waste generated by male and female-headed households in the study was plastic (45.5%). The remaining types of waste reported were paper (15.4%), glass (12.9%), food waste (10.1%), textiles (old clothes) (8.6%), and battery and electronic waste (3.9%), as seen in Table 6.

### 3.8. Reasons for Not Recycling Waste at Source

Figure 1 illustrates the reasons for not recycling waste at the source. From six options, 17.5% of the female study participants selected a lack of expertise in recycling as the most important reason for not recycling at the source, followed by a lack of awareness (15.1%), then a lack of demand for recycled products (14.3%). The lack of laws that support recycling (10.0%) was also mentioned, as well as poor infrastructure that supports recycling (7.5 %) and a lack of time (5.4%).

The reasons noted among the male study participants included poor infrastructure that supports recycling (6.8%), followed by a lack of expertise in recycling (6.1%), a lack of laws that support recycling (5.4%), a lack of awareness (4.7%), a lack of time (3.9) and a lack of demand for recycled products (3.2%).

### 3.9. Household Source of Information Related to Waste Segregation and Recycling

As shown in Table 7, there were similarities and differences among the participants’ sources of information regarding waste segregation and recycling. The majority of the male participants obtained their knowledge from social media (18.3%), followed by educational institutions (2.5%), and lastly from brochures and public meetings (2.2%). Just like the male participants, the majority of the female participants reported that their main source of information about waste segregation and recycling was social media (31.2%), followed by television (14.3%) and educational institutions (8.6%). The percentages of responses were similar and showed that social media was an important source of information about waste segregation and recycling for both the male and female respondents.

## 4. Discussion

Waste segregation and recycling at the source is a valuable tool in reducing the volume of waste and the amount of waste disposed of at landfill sites. It is also evident that waste segregation at the source is advantageous, e.g., in terms of the ease in handling, processing, enhancing resource recovery, fostering reuse and recycling, and reducing operational costs [25,26,27]. On the other hand, improper household waste solid management is associated with various health risks [21,22,23,24,27,28]. With reference to this, the authors aimed to determine households’ awareness about and practices of waste segregation and recycling. This was in addition to investigating the factors associated with households’ awareness level and practices regarding the recycling and segregation of waste.

The results of this study can be discussed from different perspectives, such as households’ practices and awareness about waste segregation and recycling, the influences of household socio-demographic variables as predictors of awareness and practices about waste segregation and recycling, reasons for households not recycling waste at the source, the association between types of waste and gender, and households’ sources of information related to waste segregation and recycling. It is noteworthy that 60.9% of the study participants had a poor overall awareness level about waste recycling. This finding is consistent with the results of other studies, which highlighted a low and inadequate level of awareness among study participants toward waste recycling in South Africa [27], Iran [29], Nigeria [30], and the UAE [31]. Moreover, the awareness level regarding waste segregation reported by the study participants was significantly lower when compared to other similar studies, such as those conducted in Afghanistan [32], Iran [33], the UAE [34], and the UK [35]. This might be a reflection of the absence of investments into waste reuse and recycling facilities, the discouragement of households to obtain more information regarding waste segregation and recycling, and a lack of an integrated program for waste management education for the public to improve their understanding of shared environmental responsibility and their participation in waste segregation and recycling [36]. This can likewise be attributed to the multi-faceted challenges related to MSW collection faced in Saudi Arabia, such as the increasing population growth, changes in habits, and a lack of awareness of the impact of solid waste on the environment [37].

Regarding the association between socio-demographic variables and awareness levels about waste segregation and recycling, it is worth noting that gender, age group, and education were associated with waste segregation and recycling. This finding is congruent with the findings of various studies, such as a study conducted in Iran, which reported that awareness about waste segregation and recycling was influenced by demographic factors such as age, education level, and gender [28]. In a study conducted in Nigeria, it was also reported that respondents’ gender and educational status had a positive impact on their awareness [38]. However, this was inconsistent with findings from another similar study conducted in Thailand [39]. Likewise, in another study conducted in Nigeria, it was highlighted that gender does not have any significant influence on awareness of household solid waste management [40]. This could be due to the social context that influences the level of waste segregation and recycling awareness.

This study revealed that family size is not associated with the level of awareness regarding waste segregation and recycling. This finding is inconsistent with the results of a study conducted in Sri Lanka [41]. These contradictory findings might be due to different family sizes, educational backgrounds, family incomes, and household cultures related to solid waste management.

The multivariate analysis in this endeavor underpins that the households with younger members of both genders, females, those with good levels of awareness about recycling, those with college and postgraduate degrees, and those who reside in urban areas were more likely to adopt good self-reported waste segregation and recycling practices at the source. This finding is collectively aligned with previous studies conducted in urban Kampala [42], Malaysia [43,44], the UK [45], Iran [46], the Czech Republic [47], and the UAE [36]. This may be linked to the Vision 2030 program, directed at the Saudi youth and meant to raise their level of responsibility toward environment.

Some scholars affirmed the fact that, indeed, women and older people are more likely to manage their waste compared to men and younger individuals. Older persons were more commonly determined to be pro-environmentalists compared to younger individuals who were shown to be less predisposed to recycling [48,49]. Women’s inclination to observe positive waste management behaviors may be attributed to the fact that traditionally, women are more focused on domestic chores, cleanliness, and health, and hence are more likely to sort waste [41,49]. In fact, there is a worldwide perception that women possess profound experience, understanding, and knowledge about their surroundings owing to their more direct and intensive interaction with the natural environment than their male counterparts [50,51]. As such, women can play significant roles throughout the waste value chain, which can vary per country. Primarily, they are consumers and disposers, and their roles further extend to being formal waste collectors, street sweepers, recycling collectors, waste bank operators, junk shop owners, or employees and factory workers.

Indeed, women can support practices toward the achievement of SDG 12 by ensuring sustainable consumption and production practices. For example, women involved in the sorting of inorganic waste can display their recycling and creative abilities to turn used materials into handicrafts, bags, and other articles which can be used personally or can be sold [52]. Correspondingly, these outcomes from various studies support the hypothesis that a household’s waste management behavior is significantly influenced by its characteristics [41,48,49].

It is noteworthy that the most common types of waste reported to be generated by households are plastic, paper, glass, food waste, textiles, and electronic waste. These findings are consistent with those of a study conducted in Malaysia regarding the waste types, but the rank differs. Food waste ranks first, followed by plastic waste [53]. Moreover, the findings are similar to those of a review conducted by Ossama et al. (2020), in which it was reported that the most common types of waste are food, paper, plastic, glass, wood, and textiles [3,54,55]. These similarities in common waste types may be due to globalization, which facilitates the spread of technology and industries to different countries where many products are produced using similar packaging methods, which leads to the production of similar types of household waste worldwide.

In the KSA, only 10–15% of this waste is recycled, while the remainder is deposited in landfill [56]. A large proportion of food waste generated in homes in large cities is reused at food banks and donated to people in need. This practice can be generalized to all recyclable waste to reduce adverse environmental effects, conserve resources, produce energy, and help build a stronger economy.

Regarding the reasons for not recycling waste at the source, in this study, we reported that the most common reasons for a female household’s failure to recycle was due to a lack of expertise, awareness, demand for recycled products, and laws that support recycling. These four reasons were included in more than half (57.0%) of the female responses, which was inconsistent with the findings of a study conducted in Bangladesh highlighting reasons for not recycling such as not having a convenient recycling scheme, a lack of time, and no space in the home [46].

Moreover, in a study in Malaysia, inadequate facilities, no time, a lack of information, too much effort required, and a lack of interest in recycling were the main reasons mentioned [36]. Furthermore, a systematic review conducted in the UK showed an inconsistent result; the reasons noted for failure to recycle included social, physical, human, economic, and policy constraints [45] and a study conducted in Nigeria showed that lack of effective community engagement is the main reason for not recycling [41]. Study participants in Kampala [43] verbalized interesting and unique reasons compared to those provided by the participants in this study and those from developing and developed countries. It is worth noting that the stated reasons were not fully related to the government but to the household’s expertise and awareness and the lack of demand for recycled products. Indeed, this is a clear call to action reflecting the growing urgency to implement collaborative initiatives by the industry and the government to further educate and increase the community’s awareness to effect desirable practices on waste recycling. Likewise, innovative technologies can be utilized to create new markets of opportunities for recyclable waste.

In terms of households’ sources of information about waste segregation and recycling, similarities and differences among the participants’ sources of information were reported. The majority highlighted social media, followed by television, and educational institutions. This result is complementary to the findings reported in studies conducted in regions such as the UAE [41]. This indicates that social media and television are effective platforms which can be utilized by public health departments at the national and local level to implement awareness campaigns to raise public awareness about waste segregation and recycling.

There are a few limitations to this study which are worth noting. The study was restricted to the head of the household, while other family members were not included. In addition, a more detailed analysis would have included a cost-benefit analysis and would have mentioned health implications. There was a non-response rate due to the fact that some households were difficult to contact, and others refused to participate. The study was limited to the Al-Ahsa province, which means that all KSA provinces would need to be studied to generalize the study’s findings. The data collection tool used was self-reports, which may have been subject to bias. Additionally, other limitations existed, such as, for example, statistical limitations in establishing causality and controlling for confounders such as waste management policies. Despite these limitations, the results of this study add to the body of knowledge and assist in future research on this topic. Moreover, our study highlighted households’ awareness and practices of waste segregation and recycling and determined the types of waste and the reasons for not segregating and recycling waste. Likewise, we aimed to identify what types of media were influential to improve public awareness about waste segregation and recycling. This information can serve as a valuable input for different stakeholders in society, such as schools, universities, and other government and private entities involved in initiatives for proper waste management systems in Al-Ahsa. This can, in turn, help the municipality to improve proper waste management processes starting at the grassroots, e.g., in residential areas. 

## 5. Conclusions and Recommendations

In this study, most of the participants had poor awareness of waste recycling and segregation. It was shown that gender, age group, and education were associated with households’ waste recycling and segregation practices. The multivariate analysis in this endeavor underpins that the younger participants of both genders and urban residents with a good level of awareness about segregation and recycling were more likely to possess good self-reported waste segregation and recycling practices at its source. The households’ most common types of reported waste were plastic, paper, glass, food waste, textiles, and electronic waste. The most common reasons for female households’ failure to recycle were a lack of expertise, awareness, demand for recycled products, and laws that support recycling. The majority highlighted social media, followed by television and educational institutions, as the household’s source of information about waste segregation and recycling.

These findings indicate a need to implement laws and develop policies to raise households’ awareness to improve the prevalence, expertise, and efficiency of recycling and segregation toward a circular economy.

The utilization of mass media and collaborations between and among the government, industries, and general education institutions and HEIs play a pivotal role in inculcating the value of caring for the environment in students’ minds, which can be implemented both in their school setting and their households. Furthermore, collaboration with these vital players in solid waste management can also be valuable in recognizing golden opportunities, initiating sustainable practices, and providing guidance to all stakeholders regarding the cultivation of the right mindset and attitudes to promote the circular economy and becoming eco-responsible citizens. In addition, inventive methods should be created to increase the demand level for recycled products such as card-based reward systems specific to a household.

## Figures and Tables

**Figure 1 ijerph-20-01790-f001:**
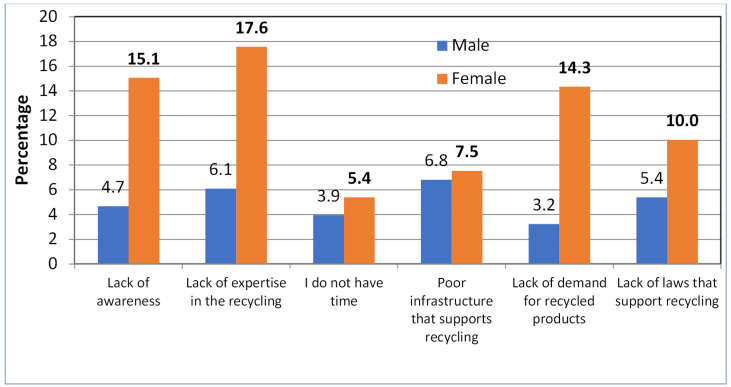
Reasons for not recycling waste at source. Chi-square, χ^2^ = 12.5, *p* = 0.024. The chi-square test result suggested significant association between gender and reasons for not recycling the waste at source (*p* = 0.024).

**Table 1 ijerph-20-01790-t001:** Socio-demographic characteristics of the study participants (*n* = 279).

Variables	Count	%
Household occupation		
- Non-governmental	39	14.0
- Governmental	89	31.9
- Self-employed	30	10.8
- Unemployed	83	29.7
- Retired	38	13.6
Household education level		
- High school and lower	46	16.5
- University College	189	67.7
- Diploma	26	9.3
- PhD	7	2.5
- Master’s degree	11	3.9
Household age group		
- 18–27 years	74	26.5
- 28–37 years	49	17.6
- 38–47 years	80	28.7
- 48–57 years	55	19.7
- >58 years	21	7.5
Homemaker education level		
- High school and lower	105	37.6
- University	97	34.8
- Diploma	18	6.5
- Postgraduate	6	2.2
- Illiterate	53	19.0
Household monthly income		
- SAR 4000–6000	21	7.5
- SAR 6001–9000	26	9.3
- SAR 9001–11,000	64	22.9
- Less than SAR 4000	27	9.7
- More than SAR 11,001	141	50.5
Gender		
- Male	84	30.1
- Female	195	69.9
Family size		
- Six members or less	144	51.6
- Seven members or more	135	48.4

Mean age of the respondents = (38.4 ± 13.1) years; average family size = (6.5 ± 2.4) SR.

**Table 2 ijerph-20-01790-t002:** Households’ variations based on their gender regarding waste management (*n* = 279).

	Variables	Male		Female		x^2^	Sign.
		No.	%	No.	%		
1	Waste considered as a problem in KSA (yes)	64	27.2	171	72.8	5.85	0.014
2	Size of waste problem in KSA (big problem)	20	25.3	59	74.7	7.45	0.045
3	Availability of waste collection services in the area (yes)	69	27.5	182	72.5	8.14	0.005
4	Frequency of waste collection in area (every day)	61	26.2	172	73.8	10.51	0.015
5	Existence of Act on waste in KSA (yes)	18	21.7	65	78.3	3.98	0.030
6	Waste management in Saudi Arabia needs more attention * (yes)	80	31.9	171	68.1	3.70	0.038

* Needs more attention from the government, the private sector, and the community.

**Table 3 ijerph-20-01790-t003:** Awareness of the households regarding waste recycling (*n* = 279).

SR	Awareness Questions/Waste Recycling	Agree	Disagree	Significance (*p*)			
		No./%	No./%	Gender	Age Group	Education	Family Size
1	Waste recycling is a shared responsibility among family members	233 (83.5)	46 (16.5)	0.404	0.232	0.001	0.05
2	Waste recycling saves money and energy	234 (83.9)	45 (16.1)	0.001	0.103	0.004	0.006
3	Waste recycling conserves the natural resources	243 (87.1)	36 (12.9)	0.047	0.936	0.706	0.013
4	Improper waste recycling is a threat to environment	211 (75.6)	68 (24.4)	0.001	0.276	0.824	0.284
5	There is need to legislate rule to encourage recycling at household level	210 (75.3)	69 (24.7)	0.025	0.05	0.305	0.733
6	By reducing the waste, we decrease the need for more landfill	228 (81.7)	51 (18.3)	0.092	0.063	0.105	0.318
7	Recycling is the process of converting waste into new products	242 (86.7)	37 (13.3)	0.228	0.61	0.7	0.009
	Overall level of awareness						
	- Poor awareness - Good awareness	170 109	60.9 39.1	0.005	0.05	0.049	0

Mean awareness score (mean ± SD: (2.17 ± 0.46) out of 3, range: (0.86–3.0)).

**Table 4 ijerph-20-01790-t004:** Awareness of the households about waste segregation (*n* = 279).

SR	Awareness Questions/Waste Segregation	Agree	Disagree	Significance (*p*)			
		No./%	No./%	Gender	Age Group	Education	Family Size
1	Individuals and community have roles and responsibilities towards waste segregation	247 (88.5)	32 (11.5)	0.011	0.001	0.023	0.967
2	Waste segregation at household level can facilitate its reuse and recycling	244 (87.5)	35 (12.5)	0.209	0.002	0.002	0.648
3	Segregation of household waste is time-consuming	148 (53.0)	131 (47.0)	0.05	0.001	0.04	0.293
4	Waste segregation containers are available near my house	55 (19.7)	224 (80.3)	0.498	0.006	0.597	0.015
5	Effective segregation of waste means that less waste is deposited in landfill	261 (93.5)	18 (6.5)	0.224	0.056	0.741	0.897
6	Overall level of awareness						
	- Poor awareness - Good awareness	196 83	70.3 29.7	0.001	0.024	0.011	0.288

Mean awareness score (mean ± SD): (1.73) out of 3, range: (1–2.60).

**Table 5 ijerph-20-01790-t005:** Logistic regression for socio-demographic characteristics of the household. Predictors of practicing waste segregation and recycling at source adjusted by level of knowledge.

Predictive Variables	Waste Recycling					Waste Segregation				
	B	Sig.	OR	95.0% C.I.		B	Sig.	OR	95.0% C.I.	
Age group										
18–27 years	R									
28–37 years	0.16	0.750	1.18	0.43	3.18	0.61	0.287	1.84	0.60	5.63
38–47 years	−1.63	0.001	0.20	0.07	0.53	0.88	0.112	2.42	0.81	7.21
48–57 years	−3.04	0.000	0.05	0.01	0.19	0.98	0.097	2.67	0.84	8.54
≥58 years	−2.91	0.001	0.05	0.01	0.31	0.73	0.338	2.08	0.47	9.31
Gender										
Male	R									
Female	1.62	0.000	5.04	2.27	11.17	1.30	0.004	2.35	1.65	2.79
Male household level of education										
High school or lower	R									
University and postgraduate	1.04	0.026	2.07	1.66	4.46	1.31	0.006	2.15	1.35	3.81
Diploma	0.21	0.793	1.24	0.25	6.12	0.78	0.251	2.19	0.57	8.36
Residence										
Alhafuf	R	0.556					0.073			
Almobaraz	−0.06	0.876	0.95	0.47	1.89	−0.33	0.329	0.72	0.37	1.40
Algorah	−0.50	0.296	0.61	0.24	1.54	−1.14	0.023	0.32	0.12	0.86
Household occupation										
Non-governmental	R	0.164					0.811			
Governmental	−0.22	0.663	0.80	0.29	2.19	0.09	0.858	1.09	0.43	2.77
Self-employed	−0.70	0.281	0.50	0.14	1.77	−0.41	0.510	0.66	0.19	2.26
Unemployed	0.74	0.187	2.10	0.70	6.33	−0.11	0.846	0.90	0.30	2.66
Retired	0.20	0.773	1.22	0.31	4.81	−0.50	0.435	0.60	0.17	2.14
Female household level of education										
High school or lower	R									
University and postgraduate	−0.27	0.473	0.76	0.36	1.60	0.71	0.062	2.04	0.97	4.31
Diploma	0.03	0.971	1.03	0.21	5.13	−1.32	0.144	0.27	0.05	1.57
Illiterate	0.82	0.092	2.28	0.87	5.92	0.43	0.351	1.54	0.62	3.82
Household Average monthly income										
SAR 4000–6000	R	0.703					0.569			
SAR 6001–9000	−0.02	0.983	0.98	0.21	4.71	−0.28	0.704	0.76	0.18	3.19
SAR 9001–11,000	0.00	0.997	1.00	0.27	3.70	−0.51	0.417	0.60	0.18	2.05
Less than SAR 4000	−0.83	0.280	0.44	0.10	1.96	−0.47	0.527	0.63	0.15	2.67
More than SAR 1001	−0.33	0.592	0.72	0.21	2.43	−0.86	0.140	0.42	0.13	1.33
Family size										
Six members or less	R									
Seven members or more	−0.42	0.235	0.65	0.33	1.32	−0.37	0.291	0.69	0.35	1.37
Waste management										
No	R									
Yes	0.13	0.774	1.14	0.47	2.79	0.14	0.761	1.15	0.48	2.74
Awareness Level										
Poor	R									
Good	1.01	0.014	2.75	1.23	6.16	1.05	0.002	2.85	1.45	5.62

**Table 6 ijerph-20-01790-t006:** Types of waste generated by households (*n* = 279).

Types of Waste	Gender				Total %	x^2^	*p*-Value
	Male		Female				
	No.	%	No.	%			
Battery and electronic waste	6	2.15	5	1.79	3.9	3.25	0.075
Textiles	4	1.43	20	7.17	8.6	2.54	0.099
Plastic	48	17.20	79	28.32	45.5	6.55	0.008
Glass	8	2.87	28	10.04	12.9	1.22	0.181
Paper	9	3.23	34	12.19	15.4	7.35	0.010
Food waste	6	2.15	22	7.9	10.1	11.14	0.020
Others	3	1.08	7	2.50	3.6	0.099	0.648
Total	84	30.11	195	69.89	100	12.8	0.046

*p*-value significant at level ≤0.05.

**Table 7 ijerph-20-01790-t007:** Household source of information related to waste segregation and recycling.

Source	Gender				x^2^	*p*-Value
	Male		Female			
	No.	%	No.	%		
Social media	51	18.3	87	31.2	14.03	0.029
Brochures	6	2.2	8	2.9	1.14	0.217
Relevant organizations	4	1.4	6	2.2	0.56	0.348
Television	5	1.8	40	14.3	9.20	0.001
Public meeting	6	2.2	12	4.3	0.95	0.470
Religious	5	1.8	18	6.5	0.84	0.255
Educational institutions	7	2.5	24	8.6	0.94	0.226
Total	84	30.1	195	69.9	14.02	0.029

The chi-square test results suggested significant association between gender and source of information related to waste segregation and recycling (*p *= 0.029). *p*-value significant at level ≤0.05.

## Data Availability

The datasets used during the current study are available with the corresponding authors and available on reasonable request.

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
