# Peer review of "Self-Reported Household Waste Recycling and Segregation Practices among Families in Eastern Region of Saudi Arabia: A Cross-Sectional Study"

_ijerph, 2023, doi:10.3390/ijerph20031790_

Round 1
Reviewer 1 Report
Find attached documents for my comments.

Author Response
Dear Reviewer,
We attach herewith our responses with regards to your valuable comments on our manuscript.

Reviewer 2 Report
The manuscript presents a study on household waste recycling and segregation practices in Saudi Arabia.
There are several studies on this topic, as the authors point out. However, it is difficult for the reader to understand why this particular manuscript is relevant to read. That is, how is it that they manage to present a different or novel point of view of this topic, in this region.
The topic of course, is always relevant, but the authors are strongly recommended to separate and use words with particular care. For example, by definition, waste recycling is the process of converting waste materials into new materials and objects, as such, that process rarely occurs at household level.
Some results (waste production) appeared out of nothing, not explanation whatsoever of of how/what units or how % were obtained from a survey, particularly because those fractions were very unlikely or at least not similar to those mentioned at the beginning of the document.
I strongly suggest an overall review of the document.
Author Response
Dear reviewr,
We attach herewith ou responses with regards to your valuable comments on our manuscript "Please see the attachment."
